# Paclitaxel Induces the Apoptosis of Prostate Cancer Cells via ROS-Mediated HIF-1α Expression

**DOI:** 10.3390/molecules27217183

**Published:** 2022-10-24

**Authors:** Yan Zhang, Yedong Tang, Xiaoqiong Tang, Yuhua Wang, Zhenghong Zhang, Hongqin Yang

**Affiliations:** 1Key Laboratory of Optoelectronic Science and Technology for Medicine of Ministry of Education, Fujian Provincial Key Laboratory for Photonics Technology, Fujian Normal University, Fuzhou 350007, China; 2Fujian Provincial Key Laboratory for Developmental Biology and Neurosciences, College of Life Sciences, Fujian Normal University, Fuzhou 350007, China

**Keywords:** paclitaxel, apoptosis, ROS, HIF-1α, JNK, prostate cancer cell

## Abstract

Prostate cancer (PCa) is the most common malignancy to endanger the health of male genitourinary system. Clinically, paclitaxel (PTX) (C47H51NO14), a diterpene alkaloid, is commonly used as an effective natural antineoplastic drug during the treatment of PCa. However, the mechanism and pathway involved in the function of PTX are poorly understood. In the current study, we employed the CCK-8 assay, revealing that PTX can inhibit the survival and induce the apoptosis of PC3M cells (a human prostate cancer cell line) in a concentration-dependent manner. Reactive oxygen species (ROS), as a metabolic intermediate produced by the mitochondrial respiratory chain, are highly accumulated under the PTX treatment, which results in a sharp decrease of the mitochondrial membrane potential in PC3M cells. Additionally, the migration and invasion of PC3M cells are weakened due to PTX treatment. Further analysis reveals that N-acetylcysteine (NAC), which functions as an antioxidant, not only rescues the decreased mitochondrial membrane potential induced by the abnormal ROS level, but also restores the migration and invasion of PC3M cells. In a subsequent exploration of the detailed mechanism, we found that hypoxia-inducible factor (HIF)-1α works as a downstream gene that can respond to the increased ROS in PC3M cells. Under PTX treatment, the expression levels of HIF-1α mRNA and protein are significantly increased, which stimulate the activation of JNK/caspase-3 signaling and promote the apoptosis of PC3M cells. In summary, we demonstrate that PTX regulates the expression of HIF-1α through increased ROS accumulation, thereby promoting the activation of JNK/caspase-3 pathway to induce the apoptosis of PCa cells. This study provides new insights into the mechanism of antineoplastic action of taxanes and unveils the clinical benefit of the ROS-HIF-1α signaling pathway, which may offer a potential therapeutic target to prevent the development of PCa.

## 1. Introduction

As the most common male genitourinary malignancy in the world, prostate cancer (PCa) is a serious threat to male health [1]. Chemotherapy is currently a clinically effective treatment for men with PCa that has spread to distant sites beyond the prostate gland, and may also be applicable to PCa patients who do not respond to hormone therapy [2]. Cytotoxic chemotherapy with taxanes is widely used for the treatment of metastatic PCa, which showed an obvious therapeutic effect. Paclitaxel, one of the most commonly used taxanes, could be used as the first-line treatment for patients with metastatic castration-resistant prostate cancer (mCRPC) [3,4]. However, the underlying mechanism of paclitaxel in the treatment of PCa has not been fully understood.

PTX is an antineoplastic agent derived from the bark of the Pacific yew tree, Taxus brevifolia. Its molecular formula is C47H51NO14, which consists of 11 chiral centers and a tetracyclic skeleton composed of 16 carbons. In cancer cells, PTX can binds to β-tubulin and prevents the dissociation of microtubules, blocking cell cycle progression, inducing mitotic arrest, then inhibiting the proliferation of cancer cells [5]. Additionally, PTX can inhibit the expression of M2 macrophage marker (CD24) by activating the Toll-like receptor 4 of macrophages and inhibiting the phosphorylation of Stat3, which leads the polarization of M2 macrophages transformed into M1, thereby killing cancer cells for the increased immunotherapy activity [6,7]. Recent reports also show that low-concentration PTX can kill cancer cells indirectly through inhibiting the proliferation of human endothelial cells and reducing the expression of angiogenesis markers, such as VEGF, TGFA and FGF, to lessen the input of nutrition [8]. Apoptosis is also one of the important ways in which PTX exerts its antineoplastic effect, but the signaling pathways involved are distinct in different cancer cells [9,10]. 

ROS are metabolic intermediates produced by the mitochondrial respiratory chain, whose intracellular level is critical for the regulation of cell proliferation and death. In cancer cells, ROS can accelerate proliferation, survival and metastasis by inducing DNA mutations and genomic instability or as a signal molecule [11,12]. However, excessive ROS will enhance oxidative stress, resulting in oxidative damage of biomolecules (protein, lipids and DNA) and inducing apoptosis or necrosis, functioning as an antineoplastic factor [13,14,15,16]. Previous studies have shown that the intracellular level of ROS in PCa cells under PTX treatment is obviously increased [17]. Hypoxia inducible factor (HIF), especially HIF-1α, is a key regulator in response to cellular stressors hypoxia, which can be induced by an excessive ROS level. Although highly expressed HIF-1α enhances tumor survival, invasion, and migration by regulating cellular metabolism in the tumor’s hypoxic microenvironment, HIF-1α also can perform as an antineoplastic factor under particular circumstances [18]. However, little is known about whether HIF-1α is involved in the antineoplastic effect of PTX, and by which mechanism HIF-1α inhibits the growth of PCa cells.

c-Jun-N-terminal kinase (JNK) is a one of the mitogen-activated protein kinases (MAPKs), which is the downstream factor of HIF-1α [19]. JNK has been shown to modulate apoptosis or proliferation of tumor cells in response to various stimuli, including cytokines, growth factors, oxidative stress, and drugs [20]. Metformin can activate JNK signaling cascade to promote cell cycle arrest and autophagy, which can be blocked by JNK inhibitor SP600125 [21]. Furthermore, siRNA-mediated knockdown of HIF-1α decreases the phosphorylation of JNK, and the expression of caspase-3 then reduces the apoptosis rate of cancer cells [22]. In cardiomyocytes, HIF-1α activated by hypoxia could induce the phosphorylation of SAPK/JNK downstream factors, like c-Jun and ATF2, resulting in efficient activation of the target genes controlled by the JNK-regulated transcription factors [19]. 

Therefore, to further explore the antineoplastic effect of PTX in PCa, and the mechanisms involved, we explored the phenotypic changes in PC3M cells under PTX treatment, and revealed that the intracellular ROS level or apoptosis rate were significantly increased compared to the control. Furthermore, the HIF-1α signaling pathway plays an essential role during this process.

## 2. Results

### 2.1. PTX Promotes the Apoptosis of PC3M Cells in a Dose-Dependent Manner

The chemical structure of PTX is presented in Figure 1A. While the structural complexity of PTX makes it an interesting and useful bioactive compound, these same features also make it difficult and expensive to prepare using totally synthetic or semi-synthetic methods. The drug concentration of nM scale has little impact on the viability of PC3M cells (Appendix A). As seen in Figure 1B, PC3M cells were treated with PTX (1–8 μM) at different doses for 24 h. Their cytotoxicity effect was measured by CCK-8 assay. Results showed that the cell viability of PC3M was decreased with the increased PTX concentration, which suggested that the cell viability of PC3M was affected by PTX in a dose-dependent manner. To confirm whether PTX could induce the PC3M cells apoptosis, we performed annexin V-FITC and PI assay. Figure 1C,D shows that PTX might induce apoptosis or necrosis of PC3M cells. It can be seen that the number of apoptotic PC3M cells was increased with the increased PTX concentration, mainly distributed at a late stage. The apoptotic percentage of the treatment with a high dose of PTX (8 μM) was up to 50%, nearly twice that of the treatment with a low dose of PTX (2 μM). In the following experiments, we further investigated the antineoplastic effect of low-dose PTX (2 μM) on PC3M cells, which also reflected the therapeutic doses given to patients [23].

### 2.2. PTX Induces ROS Generation in PC3M Cells

Reactive oxygen species (ROS), produced either from the action of NADPH oxidase (NOX) or from the mitochondrial respiratory chain, were reported to play a crucial role in progress of apoptosis [24]. The confocal images clearly depicted a significant increase in the florescence of DCFH-DA (green) of the treatment with PTX, whereas the high-dose PTX treatment (8 μM) displayed the highest florescence of DCFH-DA (green) (Figure 2A). The DCFH-DA flow cytometry assay showed that ROS levels were augmented with exposure of a higher dose of PTX in PC3M cells (Figure 2B). Thus, the above data demonstrated that, compared to the control group, PC3M cells treated with PTX showed a significant increase of ROS (Figure 2C).

### 2.3. Excessive ROS will Induce Apoptosis in PC3M Cells

The antioxidant N-acetyl cysteine (NAC), working as ROS scavenger, was applied to further analyze the elevation of ROS. We found that ROS levels activated by PTX in PC3M cells could be markedly suppressed by NAC (5 mM, 12 h) (Figure 3A,B). We then performed annexin V-FITC and PI assay, showing that the apoptotic percentage of PC3M cells with PTX (2 μM) could be markedly suppressed by NAC (5 mM, 24 h) (Figure 3C,D). The evidence showed that ROS had a role in PTX-induced apoptosis.

### 2.4. PTX Induced Apoptosis through ROS-Dependent Mitochondrial Pathway in PC3M Cells

The imbalance of ROS can promote mitochondrial dysfunction and lead to mitochondrial-mediated apoptosis [20]. This prompted us to detect the mitochondrial membrane potential (MMP) with probe JC-1, to evaluate whether mitochondrial energy production was dysfunctional in the apoptotic induction by PTX. After treatment with 2 μM PTX for 12 h, a change in fluorescence color, from red to green, was observed, showing that PTX led to a depletion of MMP in PC3M cells. Next, NAC could reverse the change of mitochondrial function induced by PTX (Figure 4A,B). Thus, the above data suggested that PTX induced PC3M cells apoptosis through ROS-dependent mitochondrial pathway.

### 2.5. PTX Inhibits the Mmigration, Invasion and Proliferation of PC3M Cells

Migration and invasion are complex processes that are involved in the malignant transformation of tumors [25]. The scratch wound assay and invasion assay were conducted to observe the cell migration of PC3M after the treatment with PTX (2 µM). PTX inhibited cell migration compared to control group at 12 h (Figure 5A,C). Similar trends were observed in the transwell invasion assay (Figure 5B,D). The number of invasive cells decreased in the treatment with PTX after 24 h. Aside from this, when compared to treatment with PTX, the number of migrated or invasive cells were slightly increased in the combination treatment with NAC and PTX (Figure 5A–D). Overall, PTX could effectively inhibit migration and invasion of PC3M cells, and ROS may play a catalytic role in inhibiting the migrated ability and invasive ability. Further to this, the PC3M-xenografted mice tumor model was developed to study the impact of the PTX treatment on prostate tumor growth. After 2 weeks of PC3M injection into mice, tumor-bearing mice were distributed into two groups (four mice in each group). One group was the control; the other was given a treatment regimen of subcutaneously injected PTX. Another 2 weeks later, tumor length and breadth were measured using a Vernier caliper. The tumor volumes of those mice treated with PTX were significantly smaller, compared to the control (Figure 5E). In conclusion, PTX effectively suppressed prostate tumor growth in vivo.

### 2.6. PTX Induces the Apoptosis through Activating ROS-Mediated HIF-1α Expression and JNK/Caspase-3 Signaling

The cellular response to hypoxia was centered on HIF-1α; we then measured the expression levels of HIF-1α. The mRNA and protein levels of HIF-1α were both increased significantly with the increase of ROS levels (Figure 6B,C). In hypoxia, HIF-1α was stabilized and mainly translocated to the nucleus, confirmed by the immunofluorescence assay (Appendix A). Furthermore, to investigate how HIF-1α control the expression of caspase-3, we measured the expression of JNK. Compared with control group, total and phosphorylated JNK protein expression were increased in the treatment with PTX, downregulated by the co-incubation with NAC and PTX (Figure 6D,E). The expression of caspase-3 was unchanged, but that of cleaved-caspase-3 was consistent with that of JNK (Figure 6F,G). We speculated that the activated JNK contributed to the release of cytochrome c from mitochondrial into the cytoplasm, leading to caspase-3 activation, which ultimately causes apoptosis (Figure 6A). Taken together, these results showed that PTX could induce apoptosis by triggering ROS-HIF-1α dependent JNK/caspase-3 cascade.

## 3. Discussion

In 2004, PTX was confirmed to prolong the overall survival (OS) of patients with mCRPC and became the first chemotherapeutic drug for the treatment of mCRPC patients in combination with androgen deprivation therapy (ADT) [26]. In this study, an castration-resistant prostate cancer cell line (PC3M) was selected to explore the antineoplastic mechanism of PTX. The cell viability of PC3M was affected by PTX in a dose-dependent manner. Interestingly, the percentage of PC3M cells apoptosis and the level of ROS were both increased, followed by the increasing concentrations of PTX. Moreover, PTX treatment inhibited the migration and invasion ability of PC3M cells. We then found that these PTX-induced phenotypic changes were significantly inhibited after NAC treatment. Aside from this, PTX also showed a satisfactory treatment effect in vivo, which attenuated the growth of tumor in BALB/c nude mice. The expression of HIF-1α was significantly increased after treating with PTX and was consistent with the changes of ROS, confirming that PTX-induced apoptosis of PC3M cells was related to ROS-mediated HIF-1α expression. 

ROS were considered to initiate carcinogenesis and support the proliferation of initiated cells during the promotion and progression stages of tumorigenesis, which may be related to the activation of GSH and TXN-mediated antioxidant system to increase the proliferative benefits [27]. The accumulation of ROS in cancer cells promoted the survival of cancer cells by blunting the activities of PTPs, PTEN and MAPK phosphatases, thereby augmenting MAPK-ERK, PI3K-PKB/Akt and PKD-NF-kB signaling cascades [28]. However, previous studies demonstrated that high ROS levels can stimulate cell death, especially during the stage of the metastatic cascade, attributed to the cytotoxicity of ROS [29]. Our results disclosed that ROS levels were stimulated by PTX, which was related to the increase of hydrogen peroxide (H_2_O_2_) [30]. Meanwhile, massive accumulation of ROS in the mitochondria could lead to apoptosis [24]. Our results showed that the MMP of PC3M cells was destroyed, which was restored by NAC. It was suggested that endogenous mitochondrial signal cascade was activated by PTX-induced ROS.

Besides, PC3M cells were considered to be a PCa line with high frequent metastasis in cancer research [31]. Our study showed that the metastasis and invasiveness of PC3M cells were significantly inhibited by PTX. It can be seen that NAC could restore the migration and invasion of PC3M cells, which may signify that massive ROS decreased the migration rate of PC3M cells, consistent with the previous study that indicated that high ROS levels are cytotoxic [11]. However, apoptosis inhibition was the main pathway that was associated with HIF-1α activation to trigger therapy resistance [32]. Under hypoxic conditions, HIF-1α exerted antioxidant effects by regulating expression of GSH-based antioxidant genes to decrease ROS production and stimulate pro-survival pathways [33]. Thus, the roles of HIF-1α in cell death was controversial. It was reported that the HIF-1α expression could lead to the deletion or mutation of p53 gene inside hypoxic cells, thus inhibiting apoptosis, which affected cellular sensitivity to chemotherapy [34]. In contrast, some data demonstrated that the accumulation of p53 protein can interact with HIF-1α subunits, so this interaction stabilized and activated P53 directly which resulted in apoptosis inside hypoxic cells [35]. Pro-apoptotic protein Nip3, act as one of the targeted gene of HIF-1α, could link to antiapoptotic Bcl-2 family members such as Bcl-2 and Bcl-XL and stimulate apoptosis by blocking these proteins [36]. Overall, many studies have suggested that HIF-1α has a critical function in the hypoxia-induced apoptosis process. Our results showed that the expression of HIF-1α protein were increased with elevated ROS levels, which is consistent with the findings of previous studies [37]. Interestingly, the mRNA levels of HIF-1α were also increased under hypoxia, indicating that the requirement for de novo transcription for HIF DNA binding under paclitaxel-induced hypoxia and the transcription levels of HIF-1α were activated [38,39]. The activated JNK/SAPK pathway has been verified to be critical for hypoxia-induced apoptosis [40]. In this study, the levels of overall JNK and p-JNK were markedly declined following HIF-1α inhibitor (NSC-13502), which proved that silencing HIF-1α was able to downregulate the activation of JNK, and HIF-1a is upstream of JNK (Appendix A). We next explored whether JNK/caspase pathway was involved in the hypoxia-induced apoptosis, which was activated by HIF-1α in PC3M cells.

JNK, known as stress-activated protein kinase, mediated the transduction of extracellular signals into cells. Previous studies reported that ursolic acid promoted apoptosis, mainly by activating JNK to mediate Bcl-2 phosphorylation and degradation, while activating caspase-9 [41]. Apoptosis induced by green tea polyphenols was mainly regulated by FAS death receptor/caspase-8 pathway, which was activated by JNK and led to apoptosis in a Fas/FasL-dependent manner [42]. In this study, it was demonstrated that the potential mechanism behind how paclitaxel affected the PC3M cells apoptosis was via ROS-mediated HIF-1α expression, as illustrated in Figure 7, which contributes new insights for a comprehensive understanding of the antineoplastic effect of HIF-1α in PCa.

## 4. Materials and Methods

### 4.1. Reagents

PTX, CCK-8 were purchased from GLPBIO (Shanghai, China). Dimethyl sulfoxide (DMSO) was obtained from Sigma-Aldrich (Shanghai, China). N-acetylcysteine (NAC), RIPA, Annexin V-FITC Detection kit and BCA Protein Assay Kit were obtained from Beyotime Institute of Biotechnology (Shanghai, China). RPMI-1640, fetal bovine serum (FBS), penicillin-streptomycin solution (PS) and phosphate-buffered saline (PBS) were obtained from Hyclone (Shanghai, China). 0.25% trypsin-EDTA was purchased from KeyGen BioTECH (Nanjing, China). DCFH-DA staining, as well as JC-1, were bought from BestBio (Shanghai, China). Transwell chamber was purchased from Corning (NY, USA). The antibodies to HIF-1α (GTX127309) were purchased from GeneTex (TX, USA). The antibodies to Phospho-JNK1/2/3 (AP0631) and JNK1/JNK3 (A5051) were purchased from abclonal (Wuhan, China). Secondary antibodies against anti-rabbit and β-actin (20536-1-AP) were bought from proteintech (Wuhan, China).

### 4.2. Cell Culture

Human prostate cancer cell line (PC3M) was purchased from the Chinese Academy of Sciences Cell Bank (Shanghai, China). Cells were cultured in RPMI-1640 supplemented with 10% FBS and 1% PS in an incubator with a 5% CO_2_ humidified atmosphere at 37 °C.

### 4.3. Cell Viability

To assess the cytotoxicity effect of PTX on PC3M cells, the CCK-8 assay was performed as per the manufacturer’s instructions. Briefly, 1.0 × 10^4^ cells were seeded in a 96-well plate and treated with DMSO (3 µL/mL), and different concentrations of PTX (1 μM, 2 μM, 4 μM, 6 μM, 8 μM) for 24 h. After incubation, 10 µL of the CCK-8 solution was added and incubated for 2h. The absorbance of cells were measured at 450 nm using a microplate reader (Bio-Rad, Hercules, CA, USA).

### 4.4. Annexin V/Propidium Iodide Staining

Briefly, PC3M cells (1.0 × 10^5^) were seeded in a 6-well plate, and after 24 h cells were treated with DMSO (3 µL/mL), different concentrations of PTX (1 μM, 2 μM, 4 μM, 6 μM, 8 μM). After 24 h, cells were trypsinized and a 195 µL annexin V Binding solution was added to each cell suspension. Then, single-cell suspensions were prepared with 5 µL of annexin V-FITC and subsequently with a 10 µL PI solution in the dark. After incubation for 20 min at room temperature, necrotic, early and late apoptosis cells were identified by flow cytometer (guava easyCyte 6-2L, Merck, NJ, USA).

### 4.5. ROS Assay

PC3M cells were seeded in a 6-well plate (1.0 × 10^5^) and in a confocal dish (2.0 × 10^4^). After 24 h, cells were exposured to DMSO (3 µL/mL), different concentrations of PTX (1 μM, 2 μM, 4 μM, 6 μM, 8 μM). One day later, cells incubated with DCFH-DA for 30 min according to the manufacturer’s introduction. Then, ROS was analyzed by flow cytometer (guava easyCyte 6-2L, Merck, NJ, USA) and observed by fluorescence microscope (Leica, Wetzlar, Germany).

### 4.6. Mitochondrial Membrane Potential (MMP) Assay

The alteration of MMP was examined by the JC-1 Assay kit. 2.0 × 10^4^ cells were seeded in a confocal dish. After 24 h, cells were treated with DMSO (3 µL/mL), NAC (5 mM), PTX (2 µM), and NAC + PTX (5 mM + 2 µM) for 12h. Then supernatants were removed from culture dishes and cells were treated with JC-1 staining solution for 20 min at 37 °C in a 5% CO_2_ incubator, and observed by fluorescence microscope (Leica, Wetzlar, Germany).

### 4.7. Wound Healing Assay

Briefly, PC3M cells were seeded in 35-mm petri dished and treated with DMSO (3 µL/mL), NAC (5 mM), PTX (2 µM), and NAC + PTX (5 mM + 2 µM). After 24 h, the viable cells were counted and then seeded in 6-well plates and grown in a monolayer. At 100% confluence, the sterile tip was used to make a straight wound. Wound photos were taken at 0 h and 12 h under a microscope (Olympus).The extent of migration was measured according to the formula ([0 h scratch width − 12 h scratch width]/[0 h scratch width]) × 100% using ImageJ software.

### 4.8. Transwell Assay

PC3M cells were seeded in 35-mm Petri dishes and treated with DMSO (3 µL/mL), NAC (5 mM), PTX (2 µM), and NAC + PTX (5 mM + 2 µM) for 24 h. PC3M cells (2.0 × 10^4^) were then counted and seeded on the upper part of the chamber. At the lower chamber, 500 µL RPMI-1640 medium with 10% FBS was added as chemoattractant. After 24 h incubation, migrated cells on the lower surface of the transwell membrane were fixed with paraformaldehyde and stained with 5% crystal violate. The number of migrated cells was counted from each well and photographed at 20× (Olympus).

### 4.9. Quantitative Real-Time RT-PCR

Total RNA was extracted from treated PC3M cells using TRIeasyTM Total RNA Extraction Reagent (YEASEN, Shanghai, China) and the mRNA was reversely transcribed into stable cDNA with Hifair^®^ III 1st Strand cDNA Synthesis Kit with gDNA digester plus (YEASEN, Shanghai, China). RT-qPCR analysis was performed by using the Hieff^®^ qPCR SYBR Green Master Mix (YEASEN, Shanghai, China). The thermos cycling conditions included: 95 °C for 5 min, followed by 40 cycles of amplification at 95 °C for 10 s and 60 °C for 30 s. GAPDH was used as an endogenous housekeeping gene. The threshold cycle (CT) was calculated in accordance with the 2^−∆∆Ct^ method relative to the expression of GAPDH. All the primer sequences are listed below: HIF-1α (forward primer: 5′ATCCATGTGACCATGAGGAAATG-3′, reverse primer: 5′-TCG GCT AGT TAG GGT ACA CTT C-3′), GAPDH (forward primer: 5′-CCA AGG CTG TGG GCA AGG-3′, reverse primer: 5′-GCT CAG TGT AGC CCA GGA TG-3′).

### 4.10. Western Blotting Analysis

PC3M cells (1.0 × 10^6^ cells/each well in 6 cm plate) were treated with DMSO (3 µL/mL), NAC (5 mM), PTX (2 µM), and NAC + PTX (5 mM + 2 µM). for 24 h. Cells were then lysed, and their proteins were isolated by using 500 µL of RIPA plus 5 µL of PMSF (Solarbio, Beijing, China). The protein concentration of each cell lysate was measured with a BCA Protein Assay Kit. 20 μg of protein was resolved by SDS–polyacrylamide gel electrophoresis (PAGE), transferred, and immunoblotted with various antibodies. HIF-1α antibody (1:1000), Phospho-JNK1/2/3 antibody (1:1000), JNK1/JNK3 (1:1000), β-actin antibody (1:3000) were used.

### 4.11. Tumor Xenograft

BALB/c nude mice were purchased from Wushi Experimental Animal Supply Co., Ltd., (Fuzhou, China) and maintained in the Laboratory Animal Center of Fujian Normal University under a 14-h light/10-h dark schedule, with a continuous supply of chow and water. PC3M cells (2.5 × 10^6^ cells/100μL) were preserved in 100 μL normal saline and injected subcutaneously into BALB/c nude mice. Once a tumor was formed, mice were randomly segregated into two different groups (four mice per group). One group was treated with DMSO; the other was given a treatment regimen with subcutaneously injected PTX. Another 2 weeks later, tumor length and breadth were measured using a Vernier caliper.

### 4.12. Statistical Analysis

All the experimental data were expressed as mean ± SD (standard deviation, *n* = 3). Statistical analysis was performed by one-way ANOVA using GraphPad Prism 6.01 software. In all cases, *p* < 0.05 was considered statistically significant.

## 5. Conclusions

In conclusion, our results uncovered the PTX antineoplastic functions on human prostate cancer with the involvement of ROS-HIF-1α dependent JNK/caspase-3 cascade. Whether HIF-1α could act as a clinical biomarker for predicting PTX response needs to be verified in subsequent studies. This work provides a useful reference and a comparative analysis for follow-up research.

## Figures and Tables

**Figure 1 molecules-27-07183-f001:**
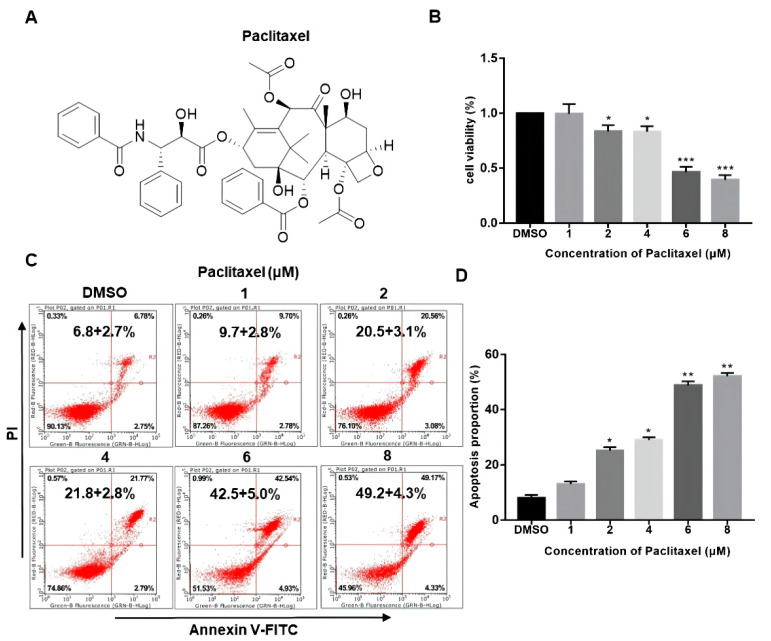
Paclitaxel inhibited the viability and promoted the apoptosis of PC3M cells. (**A**) The chemical structure of paclitaxel. (**B**) Cell viability of PC3M cells after treatment with different concentrations of paclitaxel. (**C**) PC3M cells were treated with paclitaxel (2 μM) for 24 h and stained with annexin V-FITC and propidium iodide and then analyzed by flow cytometry. (**D**) The percentage of apoptotic cells in each treatment were analyzed statistically and presented in the form of bar graphs in three independent experiments. Results are expressed as mean ± SD, *n* = 3. *: *p* < 0.01, **: *p* < 0.05, ***: *p* < 0.001.

**Figure 2 molecules-27-07183-f002:**
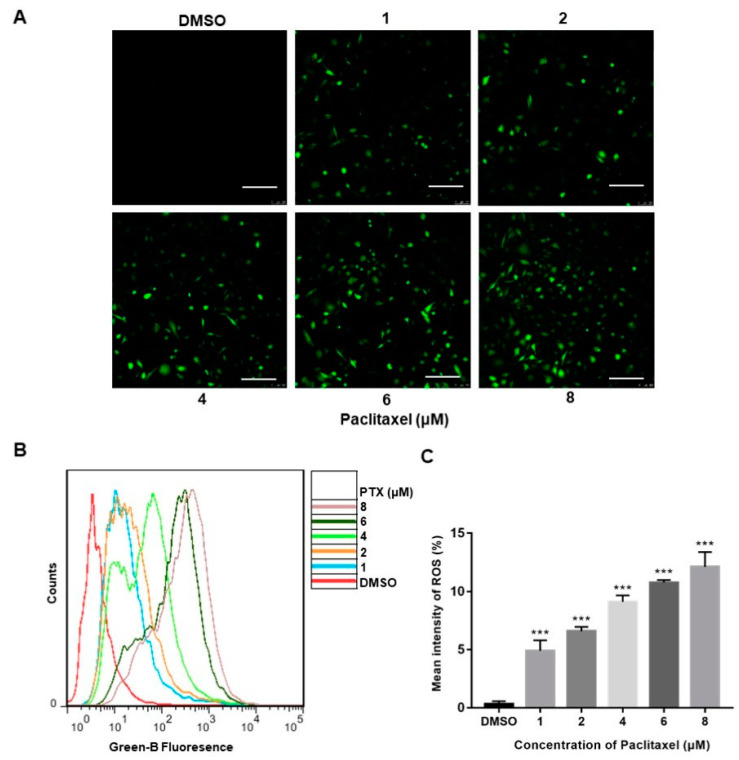
Paclitaxel-induced ROS generation in PC3M Cells. (**A**) Paclitaxel-induced ROS generation evaluated by fluorescence microscope with DCFH-DA. Scale bars = 100 μm. (**B**) ROS levels induced by paclitaxel were measured by flow cytometer. (**C**) ROS levels in each treatment were analyzed statistically and presented in the form of bar graphs in three independent experiments. Results are expressed as mean ± SD, *n* = 3. ***: *p* < 0.001.

**Figure 3 molecules-27-07183-f003:**
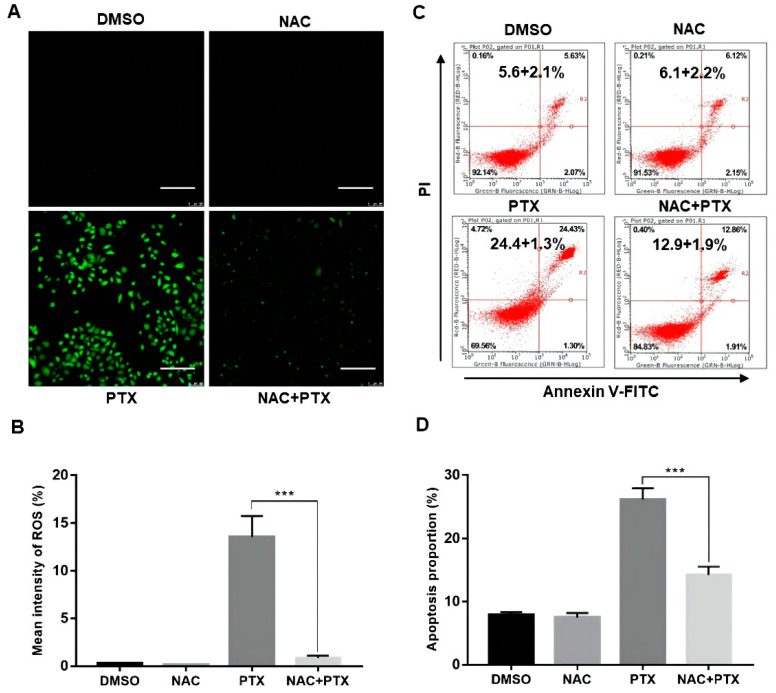
Paclitaxel induced apoptosis by inducing ROS generation in PC3M cells. (**A**,**B**) Representative images of ROS levels changes evaluated by fluorescence microscope with DCFH-DA. Scale bars = 100 μm. (**C**,**D**) NAC (5 mM) inhibited the ROS generation revealed by flow cytometric analysis. Results are expressed as mean ± SD, *n* = 3. ***: *p* < 0.001.

**Figure 4 molecules-27-07183-f004:**
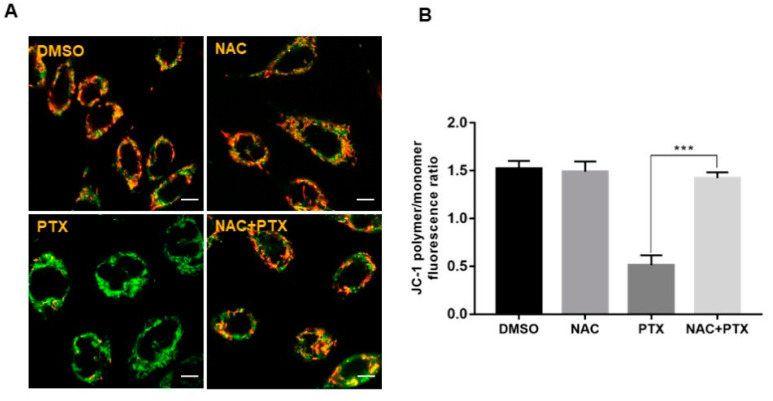
Paclitaxel induced apoptosis through ROS-dependent mitochondrial pathway in PC3M cells. (**A**,**B**) Alterations in the mitochondrial membrane after the treatment with paclitaxel were observed by JC-1 staining, following the treatment with paclitaxel with or without NAC. Scale bars = 10 μm. Results are expressed as mean ± SD, *n* = 3. ***: *p* < 0.001.

**Figure 5 molecules-27-07183-f005:**
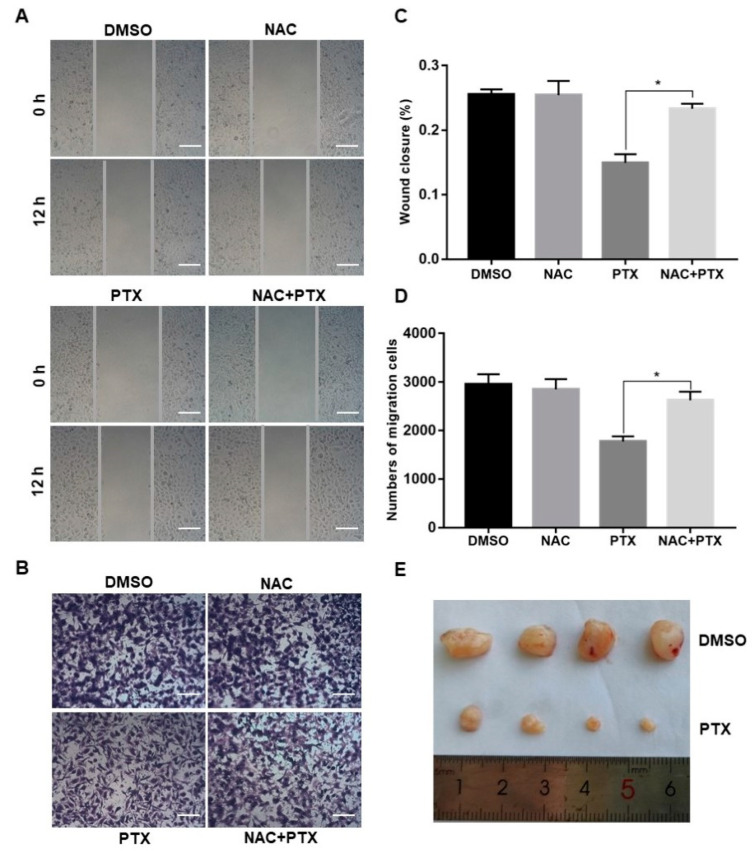
Paclitaxel inhibited cell migration and invasion in PC3M cells and suppressed prostate tumor growth in PC3M-xenografted tumor. (**A**,**C**) Cell scratch wound assay of PC3M treated with paclitaxel for 12 h. Cells were photographed at a magnification of 4× (t = 0 h, t = 12 h) and analyzed using ImageJ software. Scale bars = 50 μm. (**B**,**D**) Invasion assay of PC3M cells treated with paclitaxel for 24 h. Migrated cells were stained with crystal violet and photographed at five different areas at a magnification of 4×. Colored cells were counted and analyzed statistically. Scale bars = 50 μm. (**E**) After 2 weeks, mice were sacrificed, and tumors were isolated and photographed. Results are expressed as mean ± SD, *n* = 3. *: *p* < 0.01.

**Figure 6 molecules-27-07183-f006:**
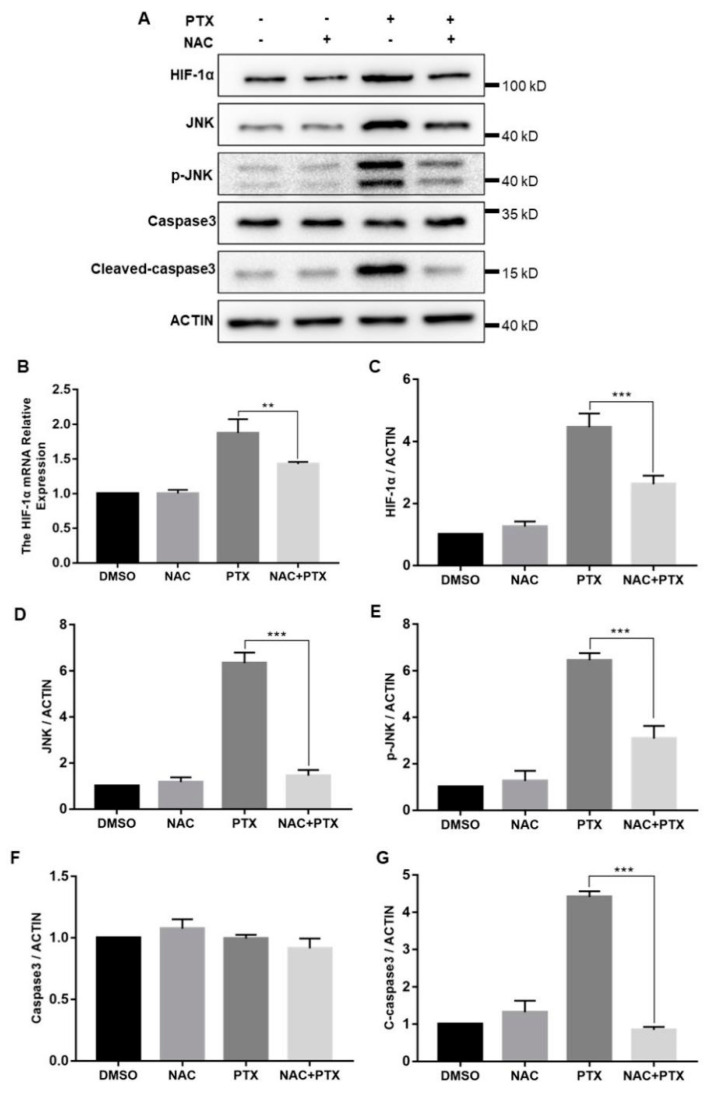
The expression of HIF-1α, JNK, P-JNK, caspase3, cleaved-caspase3 and actin in the PC3M cells were detected following the treatment with paclitaxel with or without NAC. (**A**) Representative Western blot showing the protein expression of HIF-1α, JNK, P-JNK, caspase3, cleaved-caspase3 and actin. (**B**) mRNA expression of HIF-1α was determined by real-time qPCR after the treatment with paclitaxel in presence or absence of NAC. (**C**–**G**) Summarized data of immunoblotting mentioned above. Results are expressed as mean ± SD, *n* = 3. **: *p* < 0.05, ***: *p* < 0.001.

**Figure 7 molecules-27-07183-f007:**
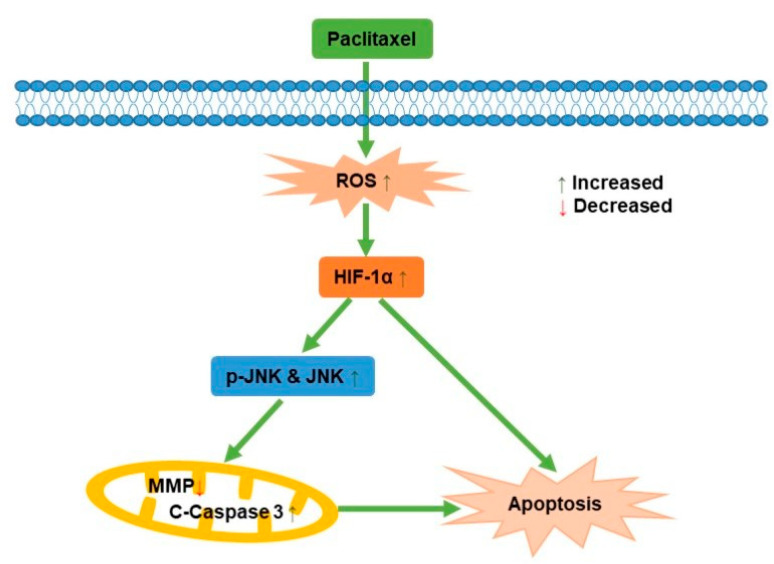
A schematic diagram for the role of paclitaxel on PC3M cell apoptosis.

## Data Availability

Data is contained within the article.

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
