# Peer review of "Paclitaxel Induces the Apoptosis of Prostate Cancer Cells via ROS-Mediated HIF-1α Expression"

_molecules, 2022, doi:10.3390/molecules27217183_

Round 1

Reviewer 1 Report

Manuscript ID: molecules-1894991

Title: Paclitaxel Induces the Apoptosis of Prostate Cancer Cells via

ROS-Mediated HIF-1a Expression

Authors: Yan Zhang, Yedong Tang, Xiaoqiong Tang, Yuhua Wang, Zhenghong Zhang, Hongqin Yang *

This study provides new insights into the antineoplastic mechanism of action of taxanes and reveals the clinical benefit of the ROS-HIF-1a signaling pathway, which may offer a potential therapeutic target to prevent the development of PCa, making it attractive; however, it would be convenient to consider the following points:

·      The expression of HIF-1a, in general, has been associated with resistance to antineoplastic treatments, so this study partially contradicts what is known. However, the experiments prove the conclusions, and it would be appropriate for the discussion to include data that contradict these results and discussion concerning this.

·      Due to the above, it would also be appropriate to demonstrate the cellular (nuclear) localization of HIF-1a with treatment and with NAC.

·      Do these doses reflect the therapeutic doses given to patients?

·      Although in the statistical analysis section it is mentioned that they were performed in triplicate, it is not mentioned how many wells were seeded for each drug concentration in cell viability or apoptosis assay; Only three wells were made, and with how many independent experiments did they make the statistics?

·      How was the NAC concentration selected for the experiments?

·      The bars on the graphs look like the standard error, not the standard deviation.

The first part of the conclusion seems very risky.

Reviewer 2 Report

The authors investigated the effect of paclitaxel in the PC3M cell line. They determined that paclitaxel causes apoptosis in these cells which is not a novel finding. They also claim an effect in migration and invasion. Furthermore, they treat mice with paclitaxel and observed smaller tumors - this is also not a novel finding. Mechanistically, the authors show that paclitaxel causes HIF-1a induction via ROS accumulation, which has been shown previously. And they try to connect HIF-1a with JNK/casp3 signaling. 

Major concerns:
1 - Are the units of paclitaxel dosage correct? uM is extremely high for this drug. The IC50 for most cells including PC3 is in the nM scale.

2 - The authors report that 2 uM of paclitaxel leads to 25% of apoptotic/dead cells. They also show that NAC treatment rescues this viability. In my view, dead cells will obviously migrate and invade less, so what the authors are measuring as lower movement of the cells, it could simply be a consequence of the fact that there are less viable cells available to migrate. 

3 - The in vivo work focuses on tumor growth, and does not address metastatic dissemination, therefore, I see no need to try to investigate invasion/migration. My advice would be to either focus on proliferation, or to investigate deeper the effects on migration by uncoupling the effect of cell dead. Perhaps the authors could sort live cells first and then do your wound healing/invasion assays. 

4 - The authors mention in the introduction that shRNA KD of HIF-1a has been shown to reduce p-JNK. They then show one western blot with increased levels of HIF-1a, p-JNK and JNK under paclitaxel treatment. The claim that HIF-1a leads to p-JNK is not supported. The western blot shows induction of overall JNK, not p-JNK particularly. Moreover, to make the claim that HIF-1a is upstream of p-JNK/JNK, the authors must employ a knockdown/knockout of HIF-1a and/or a HIF-1a inhibitor.

5 - The authors claim that "HIF-1a enhanced apoptosis via JNK/casp3". The data displayed simply does not support this conclusion. To demonstrate that HIF1-a is causing apoptosis, the authors have to first test a  knockdown/knockout of this gene, and/or a inhibitor. HIF-1a is widely known by causing chemoresistance, in particular against paclitaxel. Therefore, to really make a claim that HIF-1a is causing the opposite effect in this scenario, more data must be provided. 

6 - The authors demonstrate that ROS induces HIF-1a by showing that NAC treatment can reduce HIF-1a protein expression under paclitaxel treatment. This makes sense and it has been shown in other cell lines. The accepted mechanism by which ROS activate HIF-1a expression is by increasing HIF-1a protein stability (by reducing HIF-1a prolyl hydroxylases levels). Thus, I don't see the logic behind measuring mRNA levels of HIF1-a, and I am surprised that the authors saw an increase. 

Minor concerns:
1 - There's no point of showing fig 1a. The authors did not synthesize the compound. 

2 - The authors decided to use PC3M cells in their studies and no other cell lines. The standard of 2 cell lines to make any biological conclusion was not met. Moreover, the authors had the opportunity to compare a metastatic cell line (PCM3) with its parent (PC3) and they failed to do it. Potential differences in resistance to paclitaxel would have been interesting and novel. 

Reviewer 3 Report

Dear Authors:

The manuscript by Zhang et al has demonstrated the PTX antineoplastic functions on human prostate cancer with the involvement of ROS-HIF-1 dependent JNK/caspase-3 cascade. I have just a few suggestions.

1. Some background information or references are missing: 

In page 2, line 59-65:"ROS is metabolic intermediates produced by the mitochondrial respiratory chain, whose intracellular level is critical for the regulation of cell proliferation and death. In cancer cells, ROS can accelerate the proliferation, survival and metastasis by inducing DNA mutations and genomic instability or as a signal molecule[11]. However, excessive ROS will enhance oxidative oxidative stress, resulting in oxidative damage of biomolecules (protein, lipids and DNA) and inducing apoptosis or necrosis, function as an antineoplastic factor[12,13]." There are more reviews summarized relation between cancer development, mitochondria and ROS. (Please cite: 1. An Epigenetic Role of Mitochondria in Cancer. Cells 202211, 2518. https://doi.org/10.3390/cells11162518 

2. Advances in the Prevention and Treatment of Obesity-Driven Effects in Breast Cancers. Front Oncol. 2022 doi: 10.3389/fonc.2022.820968.

3. Mitochondrial mutations and mitoepigenetics: Focus on regulation of oxidative stress-induced responses in breast cancers. Semin Cancer Biol. 2022 Aug;83:556-569. doi: 10.1016/j.semcancer.2020.09.012.)

2. If it is possible, please add a figure to explain potential mechanism that how PTX affects cancer by involving ROS-HIF-1 dependent JNK/caspase-3 cascade.

Best,

Round 2

Reviewer 2 Report

I appreciate the responses from the authors. I leave my comments on the points that, in my opinion, still need clarification.

Points 2/3: Even though others have done similar experiments to claim effects on migration/invasion, the conclusion is still highly affected by apoptosis so I continue to believe that this experimental design does not support the reported conclusions. A simple experiment would be to pre-treat the cells, count viable cells and run the migration/invasion assay.  It's not a perfect setup but it would still be better that the current one. 

Point 4: I appreciate the use of the NSC-13502 (NSC) to demonstrate reduction of pJNK levels. Please note that total JNK is also increased with PTX and reduced with NSC, so the overall conclusion of your data should highlight JNK levels, and not necessarily phosphorylation status. 

Point 5: The ideal experimental setup to connect HIF1 to apoptosis would be to measure viability of treated cells with PTX and NCS vs PTX alone, or show the caspase 3 blot in the western added in S3. 

Point 6: Please include that comment in the discussion. 

New comment: I noticed the addition of a IF staining of HIF1a to show HIF1a in the nucleus under hypoxia. The staining of HIF1 should look 100% nuclear and there's some background in these images. But besides that, what is the point of this new data piece? It would be relevant to show nuclear staining of HIF1a with PTX treatment perhaps. 

Round 3

Reviewer 2 Report

I appreciate the efforts to provide more evidence to support the claims in the manuscript. I believe the authors covered the basics. 

Regarding the HIF-1a staining: if the finding is HIF-1a activation with PTX treatment via ROS accumulation, you should expect HIF1a stabilization and translocation to the nucleus under normal O2 conditions. Definitely in a lower extent than with the hypoxic stimulus, but still visible. You show in your WB that HIF-1a in increased, therefore stable, which should be visible in an IF staining.

Also, if this staining was performed under hypoxia, all conditions should have HIF-1a located in the nucleus. Please add a methods section for IF and this experimental setup. 

I really think this is a confusing experiment, and some clarification should be provided before publication. 
